# A Fast Calculation Method of Far-Field Intensity Distribution with Point Spread Function Convolution for High Energy Laser Propagation

**Huimin Ma [1]** , **Pengfei Zhang [2],\*** , **Jinghui Zhang [3]** , **Haiqiu Liu [1]** , **Chengyu Fan [3,4]** , **Chunhong Qiao [3,4]** , **Weiwei Zhang [1]** and **Xiaohong Li [4]**

[1] College of Information and Computer, Anhui Agriculture University, Hefei 230036, China; huiminma@ahau.edu.cn (H.M.); lhq@ahau.edu.cn (H.L.); wwzhang@ahau.edu.cn (W.Z.)
[2] Key Laboratory of High Magnetic Field and Ion Beam Physical Biology, Hefei Institutes of Physical Science, Chinese Academy of Sciences, Hefei 230031, China
[3] Key Laboratory of Atmospheric Composition and Optical Radiation, Hefei Institutes of Physical Science, Chinese Academy of Sciences, Hefei 230031, China; jhzhang@aiofm.ac.cn (J.Z.); cyfan@aiofm.ac.cn (C.F.); chqiao@aiofm.ac.cn (C.Q.)
[4] Science Island Branch, University of Science and Technology of China, Hefei 230026, China; lixiaohong@mail.ustc.edu.cn
\* Correspondence: pfzhang@aiofm.ac.cn

**Abstract:** The turbulence effect, thermal blooming effect, laser beam aberration, platform jitter, and other effects in the process of high energy laser propagation in the atmosphere will cause serious degradation of laser beam quality, which will have a negative impact on the actual application of laser propagation engineering. It is important in the engineering application of high-energy laser propagation to evaluate the far-field intensity distribution quickly. Based on the optical transfer function (OTF) theory of imaging system, the propagation process of high-energy lasers is modeled as the imaging process of point source. By using the convolution of point spread function (PSF) of jitter, turbulence, thermal blooming, and aberration of emission system, fast calculation of the far-field intensity distribution of high energy laser is realized. The calculation results are compared with those obtained by the 4D wave optics simulation program in different propagation scenarios. The results show that the calculated facula distribution and encircled energy of this method are in good agreement with the simulation results of wave optics, which can realize the fast and accurate evaluation of the far-field intensity distribution of high-energy laser propagation and provide a reference for practical engineering application.

**Keywords:** atmospheric optics; high energy laser; turbulence; thermal blooming; convolution

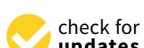



## 1. Introduction

A series of propagation effects, such as turbulence effect and thermal blooming effect, will be produced in the process of high energy laser (HEL) propagation in the atmosphere [1]. Many works have verified that these propagation effects, laser beam aberration, and platform jitter will cause serious degradation of laser beam quality, which will bring adverse effects to the actual application of laser propagation engineering [2–6]. In the engineering application of high-energy lasers, the beam quality is an important metric. The rapid evaluation of the beam quality of laser far-field long exposure can provide an important reference for the practice and optimization of laser systems.

At present, researchers generally use the method of scaling law to analyze the beam quality of laser far-field long exposure facula; that is, through a 4D wave optics program to simulate the far-field peak intensity and the beam expansion radius under a certain ring energy under different turbulence, thermal blooming, initial beam quality, and other parameters, and use the characteristic parameters describing these effects to fit the results to

achieve the characteristics parameter scaling of far-field beam quality. In 1976, Gebhardt [7] obtained the expression of far-field facula peak power of a Gaussian beam and a truncated Gaussian beam by using the generalized thermal distortion parameter, and obtained the scaling formula of laser far-field facula expansion based on the RMS approximation theory, assuming that the far-field distribution is Gaussian distribution. In 1977, Smith et al. [8] presented the expression of the peak power of the far-field facula of the platform beam laser, but lacked the scaling results of the spread of the platform beam facula. Subsequently, Y. Huang [9] obtained the scaling law of the far-field facula expansion of the platform beam. Based on the Bradley Herrmann thermal distortion parameter [10], R.Stock [11] obtained the scaling relationship of the facula spread under the combined effect of the turbulence thermal of the platform beam. In 2006, Ngwele [12] found that, under the same thermal distortion parameters, the peak intensity of laser up-line and down-line propagation was significantly different, and the influence of the thermal blooming phase screen at different positions in the propagation path on the peak value of far-field facula was also different. By introducing propagation scaling and path weight function to improve the thermal distortion parameters, Ngwele could reduce the dependence of the Fresnel number and extinction path change in the scaling relationship. In 2008, N. Scott [13] modified the thermal distortion parameter to make it suitable for the evaluation of the defocused laser propagation effect. In 2010, C. Qiao [14,15] obtained the scaling rule of laser atmospheric horizontal propagation at the 1 μm and 4 μm band by using this parameter, and analyzed the improved method of thermal distortion parameter $N_d$ under different turbulence effects [14–16], pointing out that, when the turbulence effect is weak, it is assumed that the turbulence and thermal blooming effect are independent; when the turbulence effect is strong, the thermal blooming effect acts on the basis of turbulence expansion.

Based on the above study of the scaling law, we can summarize that there are some limitations in the evaluation of laser far-field facula with the scaling law. First, due to the difference of the diffraction effect of the facula shape and the difference of the propagation scene, different scaling relations are needed for different turbulence conditions, the initial light field distribution, and the propagation scene. Second, the scaling law is only for the expansion of facula under the specific normalized annular energy in the scaling; if the normalized energy of the ring is changed, the fitting form of the scaling law will also be changed. Third, in the evaluation of the far-field beam quality, the scaling law assumes that the facula is a Gaussian distribution, which cannot accurately describe the far-field beam distribution.

In recent years, many researchers are actively exploring new methods to calculate the far-field intensity distribution on HEL propagation. In 2016, A. Sami [17] obtained the scaling rule of arbitrary aperture shape, beam type, and array beam under vacuum condition. However, P. Bingham [18] found that the formula was not suitable for beam quality evaluation under turbulent conditions through wave optics numerical simulation in 2018. R. Van Zandt [19] found that the spot distribution under the combined effect of turbulence and thermal blooming can be obtained by convoluting the far-field spot image of the simple turbulence effect and the far-field spot image of the simple thermal blooming effect. Compared with the simulation of wave optics, this method is a new HEL scaling law model for fast, accurate, and enhanced modeling of combined thermal blooming and turbulence effects on HEL propagation. However, Zandt's method is an empirical exploration and at present it lacks a theoretical basis, as well as a more comprehensive analysis considering platform jitter, initial aberration of the laser system, and other factors.

If we regard the transmission channel of the laser as an optical system, the calculation of the spot shape after the laser is transmitted through the atmosphere can be regarded as the calculation of the PSF of the system. In the field of astronomical imaging and adaptive optics (AO), the PSF calculation is extremely important and needs to be accurately known because it provides crucial information about optical systems for design, characterization, and optical diagnostics. It is generally difficult to obtain an accurate PSF. Scholars generally use empirical, theoretical, or parameter fitting methods, e.g., E. Steinbring [20] proposed a

semi-empirical technique that can improve the determination of AO off-axis PSF. The results show that this simple method reduces error in the prediction of the basic radial variation in the PSF. M. C. Britton [21] derived an analytic formulation of the Anisoplanatic Point-Spread Function. The analytic formulation can captures the dependencies of anisoplanatism on aperture diameter, observing wavelength, angular offset, zenith angle, and turbulence profile. R. JL. Fétick [22] developed a model of the AO long-exposure PSF based on end-to-end simulated PSFs using the OOMAO. S. A. Shakir [23] has proposed that the irradiance of partially coherent light propagating under the influence of multiple random effects is the convolution of irradiance propagating in vacuum and the point spread function of the system representing random effects.

Based on R. Van Zandt's new scaling law model results and the calculation method of PSF proposed by scholars, in this paper we try to explore and validate the theory model of the convolution method for HEL propagation with turbulence and thermal blooming effects. In this study, the process of laser propagating to the target is equivalent to the imaging process of point source, which provides theoretical support for the method. At the same time, considering the aberration of laser beam and the jitter effect of the platform, the method is extended. Finally, to test our theory and model, the improved method is compared with the simulation results of a four-dimensional program of wave optics.

## 2. Theory and Model

### 2.1. Fast Calculation of Far-Field Intensity Distribution with PSF Convolution

In our model assumptions, the propagation process of the laser from emission system to target can also be regarded as an imaging process of infinite point light source, and the schematic diagram is shown in Figure 1. Far-field intensity can be obtained by calculating the *PSF* of the point light source imaging system, in which a series of effects, such as limited aperture, platform jitter, atmospheric turbulence, and thermal blooming are considered.

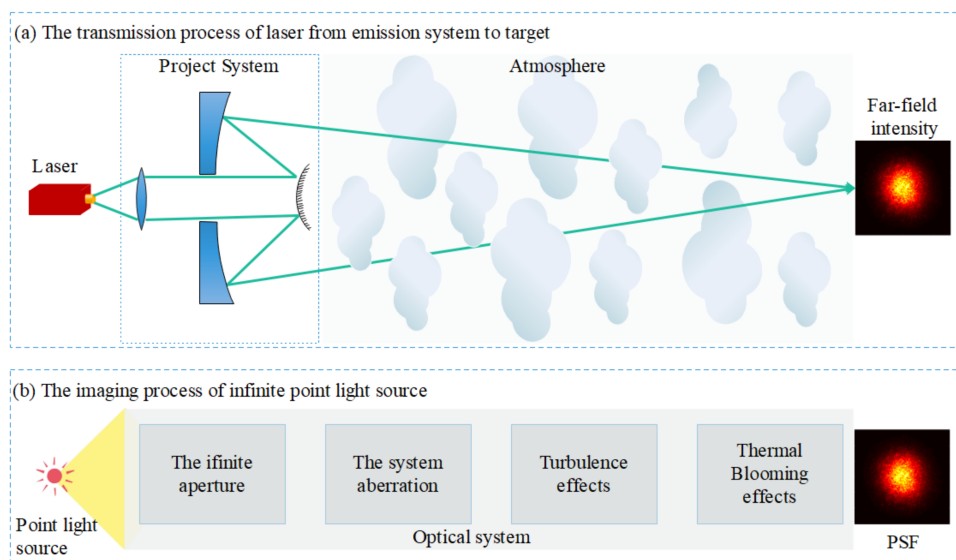

**Figure 1.** Schematic diagram for (**a**) the propagation process of laser from emission system to target and (**b**) the imaging process of infinite point light source.

Considering the finite aperture and aberration of the laser emission system, the *OTF* of the imaging system is the normalized autocorrelation function of the generalized pupil function and can be expressed as [24]:

$$OTF(\lambda z f_x, \lambda z f_y) = \frac{\int \int_{-\infty}^{\infty} p(x,y) p(x-\lambda z f_x, y-\lambda z f_y) exp(i\psi(x,y)) exp\left(-i\psi(x-\lambda z f_x, y-\lambda z f_y)\right) dx dy}{\int \int_{-\infty}^{\infty} p^2(x,y) dx dy} \tag{1}$$

where $p(\lambda z f_x, \lambda z f_y)$ is aperture function, $\lambda$ is laser wavelength, $z$ is focal length, $f_x, f_y$ are the coordinates of spatial frequency, and $\psi(\lambda z f_x, \lambda z f_y)$ is the aberration function of the aperture. When turbulence, jitter, and other random phase disturbances are considered, $OTF_L$ with long exposure, can be expressed as the ensemble average of the random phase,

$$
\begin{gathered}
OTF_L(\lambda z f_x, \lambda z f_y) = \\
\frac{\int\int_{-\infty}^{\infty} p(x,y)p(x-\lambda z f_x, y-\lambda z f_y)\langle exp(i\psi(x,y))exp\left(-i\psi(x-\lambda z f_x, y-\lambda z f_y)\right)\rangle dxdy}{\int\int_{-\infty}^{\infty} p^2(x,y)dxdy}
\end{gathered}
\tag{2}
$$

$\psi$ is expressed as the sum of turbulence effect, $\psi_{turbulence}$; jitter, $\psi_{jitter}$; thermal blooming effect, $\psi_{TB}(x,y)$; aperture of launch system; and initial aberration of system, $\psi_p(x,y)$, as follows:

$$
\psi(x,y) = \psi_{turbulence}(x,y) + \psi_{jitter}(x,y) + \psi_{TB}(x,y) + \psi_p(x,y)
\tag{3}
$$

It is assumed that the phase of turbulence effect, thermal blooming effect, and jitter are independent of each other, and that the aberration caused by the thermal blooming effect will not change when the thermal blooming effect of the CW laser reaches steady state. In this case, the thermal blooming effect is considered as the initial aberration of the aperture $\psi_{p+TB}(x,y) = \psi_{TB}(x,y) + \psi_p(x,y)$. Considering the phase disturbances above, the aberration function can be expanded to:

$$
\begin{gathered}
\langle exp(i\psi(x,y))exp\left(-i\psi(x - \lambda z f_x, y - \lambda z f_y)\right)\rangle \\
= \langle \exp\left(i\psi_{jitter}(x,y)\right) \exp\left(-i\psi_{jitter}(x - \lambda z f_x, y - \lambda z f_y)\right)\rangle \\
\times \langle \exp(i\psi_{turbulence}(x,y)) \exp\left(-i\psi_{turbulence}(x - \lambda z f_x, y - \lambda z f_y)\right)\rangle \\
\times exp\left(i\psi_{p+TB}(x,y)\right)exp\left(-i\psi_{p+TB}(x - \lambda z f_x, y - \lambda z f_y)\right)
\end{gathered}
\tag{4}
$$

Furthermore, if the turbulence and jitter phase are generalized stationary, then the ensemble mean value of the jitter effect and turbulence effect is independent of the $x$, $y$ coordinates, and moves it outside the integral, then Equation (2) can be expressed as:

$$
\begin{gathered}
OTF_L(\lambda z f_x, \lambda z f_y) \\
= \langle \exp, \left(i\psi_{jitter}(x,y)\right), \exp, \left(-i\psi_{jitter}(x - \lambda z f_x, y - \lambda z f_y)\right)\rangle \\
\times \langle \exp(i\psi_{turbulence}(x,y)) \exp\left(-i\psi_{turbulence}(x - \lambda z f_x, y - \lambda z f_y)\right)\rangle \\
\times \frac{\int\int_{-\infty}^{\infty} p(x,y)p(x-\lambda z f_x, y-\lambda z f_y)exp\left(i\psi_{p+TB}(x,y)\right)exp\left(-i\psi_{p+TB}(x-\lambda z f_x, y-\lambda z f_y)\right)dxdy}{\int\int_{-\infty}^{\infty} p^2(x,y)dxdy} \\
= OTF_{jitter}(\lambda z f_x, \lambda z f_y)OTF_{turbulence}(\lambda z f_x, \lambda z f_y)OTF_{p+TB}(\lambda z f_x, \lambda z f_y)
\end{gathered}
\tag{5}
$$

where $OTF_{jitter}$, $OTF_{turbulence}$, and $OTF_{p+TB}$ are, respectively, the *OTF* for jitter effect, turbulence effect, and thermal blooming effect with aberrations and finite aperture effect. In the actual calculation process, it is convenient to use the inverse Fourier transform of *OTF*, that is, to calculate the *PSF* in the spatial domain. In the spatial domain, the system *PSF* can be expressed in the form of convolution:

$$
PSF_{FC}(x,y) = PSF_{jitter}(x,y) \otimes PSF_{turbulence}(x,y) \otimes PSF_{p+TB}(x,y)
\tag{6}
$$

where $PSF_{FC}$ is the system *PSF* of the fast calculation model with *PSF* convolution, $PSF_{jitter}$, $PSF_{turbulence}$, and $PSF_{p+TB}$ is, respectively, the *PSF* for jitter effect, turbulence effect, and thermal blooming effect with aberrations and finite aperture effect, and $\otimes$ represents convolution operator.

### 2.2. PSF Calculation for Jitter Effect

The system jitter is modeled as a random tilt phase screen, which can be expressed as:

$$
\psi_{jitter}(x,y) = \exp\left(ik(\theta_x x + \theta_y y)\right)
\tag{7}
$$

where $\theta_x, \theta_y$ are the angles between the optical axis direction and $x, y$ direction, respectively, $k$ is wave vector, and $i$ is imaginary unit. Considering that $\theta_x, \theta_y$ is a normal distribution with a mean value of 0 and a root mean square of $\theta$, then the *OTF* of the jitter effect can be expressed as:

$$
\begin{aligned}
OTF_{jitter} &= \langle \exp(i\psi_{jitter}(x_1,y_1))exp(-i\psi_{jitter}(x_2,y_2))\rangle \big|_{\Delta x=\lambda z f_x, \Delta y=\lambda z f_y} \\
&= \exp\left(-\tfrac{1}{2}\overline{(\phi(x_1,y_1)-\phi(x_2,y_2))^2}\right)\big|_{\Delta x=\lambda z f_x, \Delta y=\lambda z f_y} \\
&= \exp\left(-\tfrac{1}{2}k^2\theta^2\Delta r^2\right)\big|_{\Delta r=\lambda z f} \\
&= \exp\left(-2\pi^2\theta^2 z^2 f^2\right)
\end{aligned}
\tag{8}
$$

where $\Delta r = \sqrt{(x_2-x_1)^2+(y_2-y_1)^2}$, $\Delta x = x_2-x_1$, $\Delta y = y_2-y_1$, and $f = \sqrt{f_x^2+f_y^2}$. The corresponding PSF in the spatial domain is:

$$
PSF_{jitter} = \exp\left(-\frac{r^2}{2\theta^2 z^2}\right)
\tag{9}
$$

*2.3. PSF Calculation for Turbulence Effect*

The *OTF* for turbulence effect [25,26] can be expressed as

$$
\begin{aligned}
OTF_{turbulence} &= \langle \exp(i\psi_{turbulence}(x_1,y_1))exp(-i\psi_{turbulence}(x_2,y_2))\rangle \big|_{\Delta x=\lambda z f_x, \Delta y=\lambda z f_y} \\
&= \exp\left(-\tfrac{1}{2}D_\phi(\lambda z f)\right)
\end{aligned}
\tag{10}
$$

where $D_\phi(\lambda z f) = 6.88\left(\frac{\lambda z f}{r_0}\right)^{5/3}$ and $r_0$ is the Fried parameter. The corresponding *PSF* in the spatial domain is:

$$
PSF_{turbulence} = F^{-1}\left(\exp\left(-3.44\left(\frac{\lambda z f}{r_0}\right)^{5/3}\right)\right)
\tag{11}
$$

where $F^{-1}$ represents the inverse Fourier transform operation.

*2.4. PSF Calculation for Thermal Blooming Effect with Aberrations and Finite Aperture Effect*

The *OTF* for thermal blooming effect with aberrations and finite aperture effect is expressed as:

$$
\begin{aligned}
&OTF_{p+TB}(\lambda z f_x, \lambda z f_y) \\
&= \frac{\int\int_{-\infty}^{\infty} p(x,y)p(x-\lambda z f_x, y-\lambda z f_y)\exp\left(ik\psi_{p+TB}(x,y)\right)\exp\left(-ik\psi_{p+TB}(x-\lambda z f_x, y-\lambda z f_y)\right)dxdy}{\int\int_{-\infty}^{\infty} p^2(x,y)dxdy}
\end{aligned}
\tag{12}
$$

In general, Equation (12) has no analytical solution. When the thermal blooming effect is weak and can be ignored, the *PSF* corresponding to aperture and system aberration can be calculated by the modulus square of the Fourier transform of the complex function composed of aperture function and aberration. The thermal blooming effect can be calculated quickly by the method of AOTB [19]. In order to verify the validity of the transfer function method, a 4D wave optics simulation program is still used to calculate the *PSF* of the thermal blooming effect.

## 3. Results and Discussion

*3.1. Calculation Parameter Setting*

In order to verify the applicability of the fast calculation model to a wide range of practical applications, we apply $PSF_{FC}$ for a fast calculation model to the multiple high-power laser propagation scenarios, which are designed to encounter various effects,

including the turbulence effect, the thermal blooming effect, laser beam aberration, and platform jitter.

The emitting laser is a flatform beam that is horizontally focused on the target. We modeled HEL propagation of the laser beam through turbulence and thermal blooming as a sequence of 2D thin phase screens, and the wave propagation from one thin phase screen to another using scalar diffraction theory [27,28]. The fields propagated between the phase-screens were calculated by a sequence of the numerically efficient fast-Fourier-transform (FFT) operations [29,30]. The random turbulence phase screens were generated with von Karman spectrum distributions by white noise filtering in the Fourier domain. The refractive index structure parameter of atmospheric turbulence near the surface is affected by underlying surface and weather conditions, and its variation range is usually in the range of $10^{-15}$ m$^{-2/3}$ and $10^{-13}$ m$^{-2/3}$ [31,32]. We selected three typical turbulence refractive index structure parameters to simulate different atmospheric turbulence conditions. The optical turbulence strength of our simulated is given in detail in Table 1. Forty independent statistical turbulence-induced phase screens were laid over the propagation path. Non-adaptive coordinate transformation [29,33] was adopted to enhance both the speed and the accuracy of the computation. The density changes of thermal blooming was calculated with a time-dependent hydrodynamic model by finite Fourier series method [34]. The system and propagation parameters of interest in the 4D wave optics simulation are described in Table 1, and the scenarios are defined in Table 2, as follows.

**Table 1.** Parameters of interest in the 4D wave optics simulation.

| Parameter | Value(s) |
|---|---|
| Grid number, N | $1024 \times 1024$ |
| Grid sampling interval of the emission plane, $\Delta x$ | 0.0012 m |
| Number of phase screen, $N_{ps}$ | 40 |
| Wavelength, $\lambda$ | 1 μm |
| Transmitting aperture, D | 0.3 m |
| Propagation distance, L | 3 km |
| Initial aberration of the laser emission system, $\beta_0$ | 1.0 and 5.0 times diffraction limit aberration |
| Turbulence structure constant, $C_n{}^2$ | $5.0 \times 10^{-14}$ m$^{-2/3}$, $1.0 \times 10^{-14}$ m$^{-2/3}$, $5.0 \times 10^{-15}$ m$^{-2/3}$ |
| Absorption coefficient, $\alpha$ | $1.1 \times 10^{-5}$ m$^{-1}$ |
| Extinction coefficient, $\xi$ | $6.1 \times 10^{-5}$ m$^{-1}$ |
| Wind speed, v | 2 m/s |
| Angular velocity of light beam scanning, $\omega$ | 0 rad/s |

**Table 2.** Scenarios used in the wave optics simulations.

| Scenario | $C_n{}^2$ | $D/r_0$ | jitter | $\beta_0$ | P | $N_D$ |
|---|---|---|---|---|---|---|
| S1 | $5.0 \times 10^{-15}$ m$^{-2/3}$ | 4.0 | 0 | 1 | - | - |
| S2 | $5.0 \times 10^{-14}$ m$^{-2/3}$ | 16.1 | 0 | 1 | - | - |
| S3 | $5.0 \times 10^{-15}$ m$^{-2/3}$ | 4.0 | 5 μrad | 5 | - | - |
| S4 | $5.0 \times 10^{-14}$ m$^{-2/3}$ | 16.1 | 10 μrad | 5 | - | - |
| S5 | $1.0 \times 10^{-14}$ m$^{-2/3}$ | 6.13 | 5 μrad | 5 | 10 kW | 25.5 |
| S6 | $1.0 \times 10^{-14}$ m$^{-2/3}$ | 6.13 | 5 μrad | 5 | 50 kW | 127.3 |
| S7 | $1.0 \times 10^{-14}$ m$^{-2/3}$ | 6.13 | 5 μrad | 5 | 100 kW | 254.3 |

The strength of turbulence effect is usually measured by atmospheric coherence length (Fried parameter) $r_0$. For a spherical wave and an uplink path, when the zenith angle is 0, $r_0$ is calculated using the formula [26]:

$$r_0 = \left[ 0.423 k^2 \int_0^L C_n^2(z)(1 - z/L)^{5/3} dz \right]^{-3/5} \tag{13}$$

where $k = 2\pi/\lambda$ is wave number, $C_n^2(z)$ is the turbulent refractive structure parameter at position $z$, and $L$ is the transmission distance. In the case of a horizontal path in which the refractive index structure parameter, $C_n^2$, is assumed as a constant, Fried's parameter can be expressed as $r_0 = (0.16C_n^2k^2L)^{-3/5}$. The intensity of the thermal blooming effect is described by the Bradley–Herrman thermal distortion parameter, $N_D$ [10], which is defined as:

$$N_D = 4\pi\sqrt{2}C_0\lambda^{-1}\int_0^R \{\alpha(z)(P\exp(-\int_0^z dz'\xi(z')))/[|\vec{v} + z\vec{\omega}|D(z)]\}dz, \qquad (14)$$

where $C_0 = 1.66 \times 10^{-9}$ m$^3$/J, $P$ is the laser emission power, $\alpha$ is the absorption coefficient, $v$ is the wind speed along the propagation path, $\omega$ is the angular velocity of light beam scanning, and $\xi$ is the atmospheric extinction coefficient. Results of long-exposure average can help to reveal the statistical average effects of spot position wandering and spreading on the overall energy spread. The long-exposure results in the simulation were averaged over an ensemble of 60 statistical independent random realizations.

### 3.2. PSF Calculation for Different Effects

According to fast calculation using the convolution we proposed in Equation (6), we first calculates $PSF_{jitter}$, $PSF_{Turbulence}$, and $PSF_{p+TB}$, for the single effect. The $PSF$ under different jitter conditions, $PSF_{jitter}$, is calculated according to Equation (9). Figure 2 shows the results for *jitter* = 5 *μrad* and *jitter* = 10 *μrad*, which are recorded as $PSF_{jitter=5\mu rad}$ and $PSF_{jitter=10\mu rad}$, respectively, in the article.

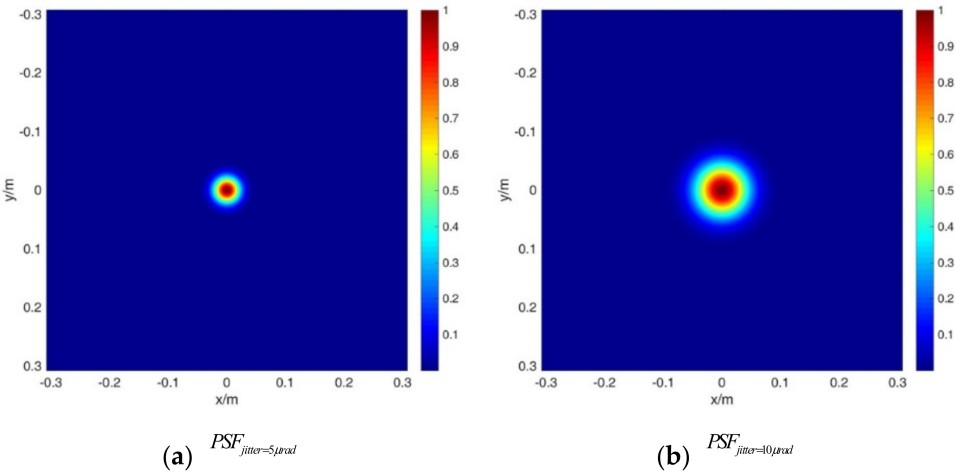

(a) $PSF_{jitter=5\mu rad}$　　　　　　　　　　　　(b) $PSF_{jitter=10\mu rad}$

**Figure 2.** The *PSF* results for (**a**) *jitter* = 5 *μrad* and (**b**) *jitter* = 10 *μrad*.

The *PSF* under different turbulence conditions, $PSF_{tuebulence}$, is calculated according to Equation (11). Figure 3 shows the results for $D/r_0 = 4.0$, $D/r_0 = 6.13$ and $D/r_0 = 16.1$, which are recorded as $PSF_{D/r_0=4.0}$, $PSF_{D/r_0=6.13}$, and $PSF_{D/r_0=16.1}$, respectively.

Figure 4 shows the corresponding *PSF* of the transmitting system without and with aberration, which are recorded as $PSF_{\beta_0=1}$ and $PSF_{\beta_0=5}$, respectively.

Figure 5 shows the *PSF* of the system without and with aberration under different thermal distortion parameters. The PSF is calculated by a 4D wave optics simulation program, and is recorded as $PSF_{\beta_0=1,N_D=25.5}$, $PSF_{\beta_0=1,N_D=127.3}$, $PSF_{\beta_0=1,N_D=254.3}$, $PSF_{\beta_0=5,N_D=25.5}$, $PSF_{\beta_0=5,N_D=127.3}$, and $PSF_{\beta_0=5,N_D=254.3}$, respectively.

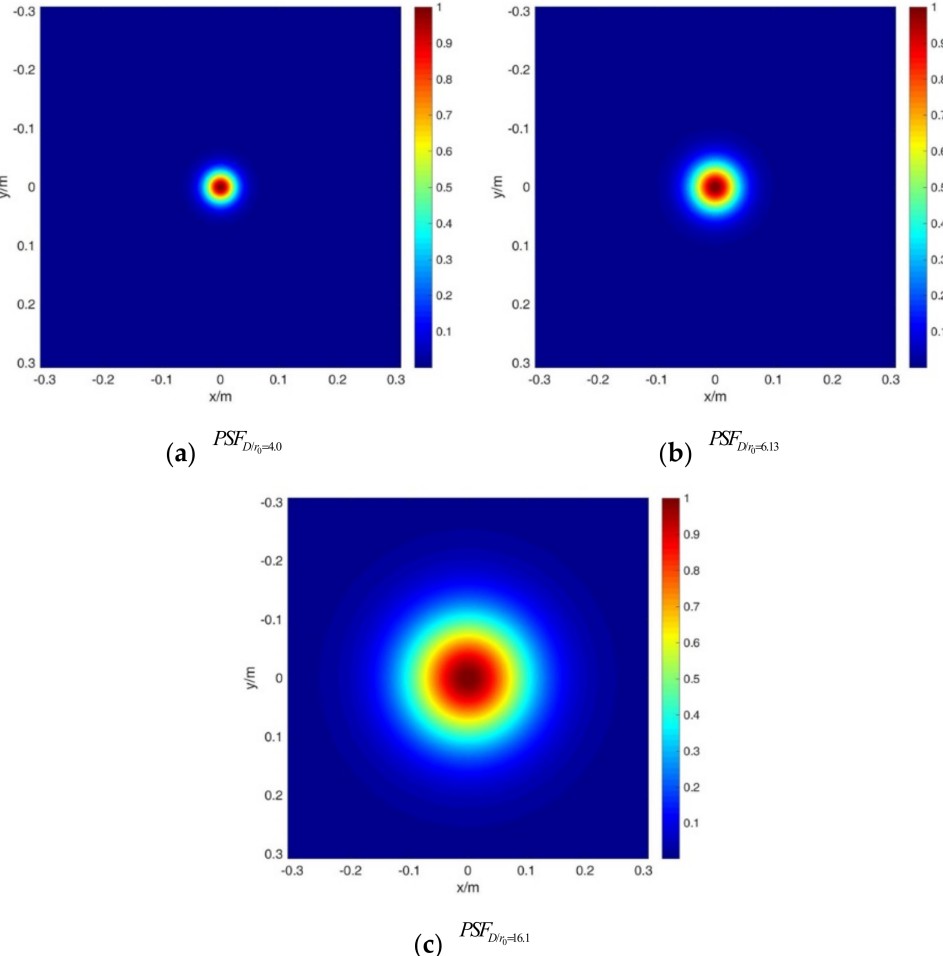

**Figure 3.** The *PSF* results for (**a**) $D/r_0 = 4.0$, (**b**) $D/r_0 = 6.13$, and (**c**) $D/r_0 = 16.1$.

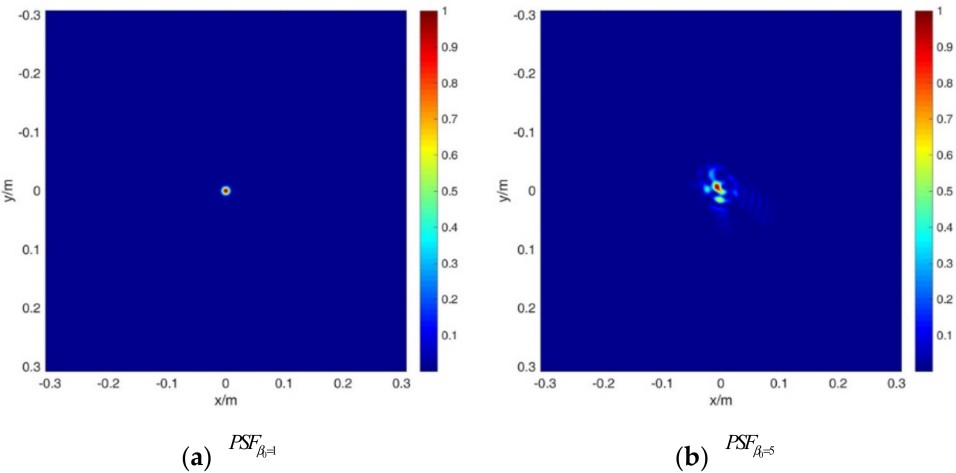

**Figure 4.** The *PSF* results for (**a**) $\beta_0 = 1$ and (**b**) $\beta_0 = 5$.

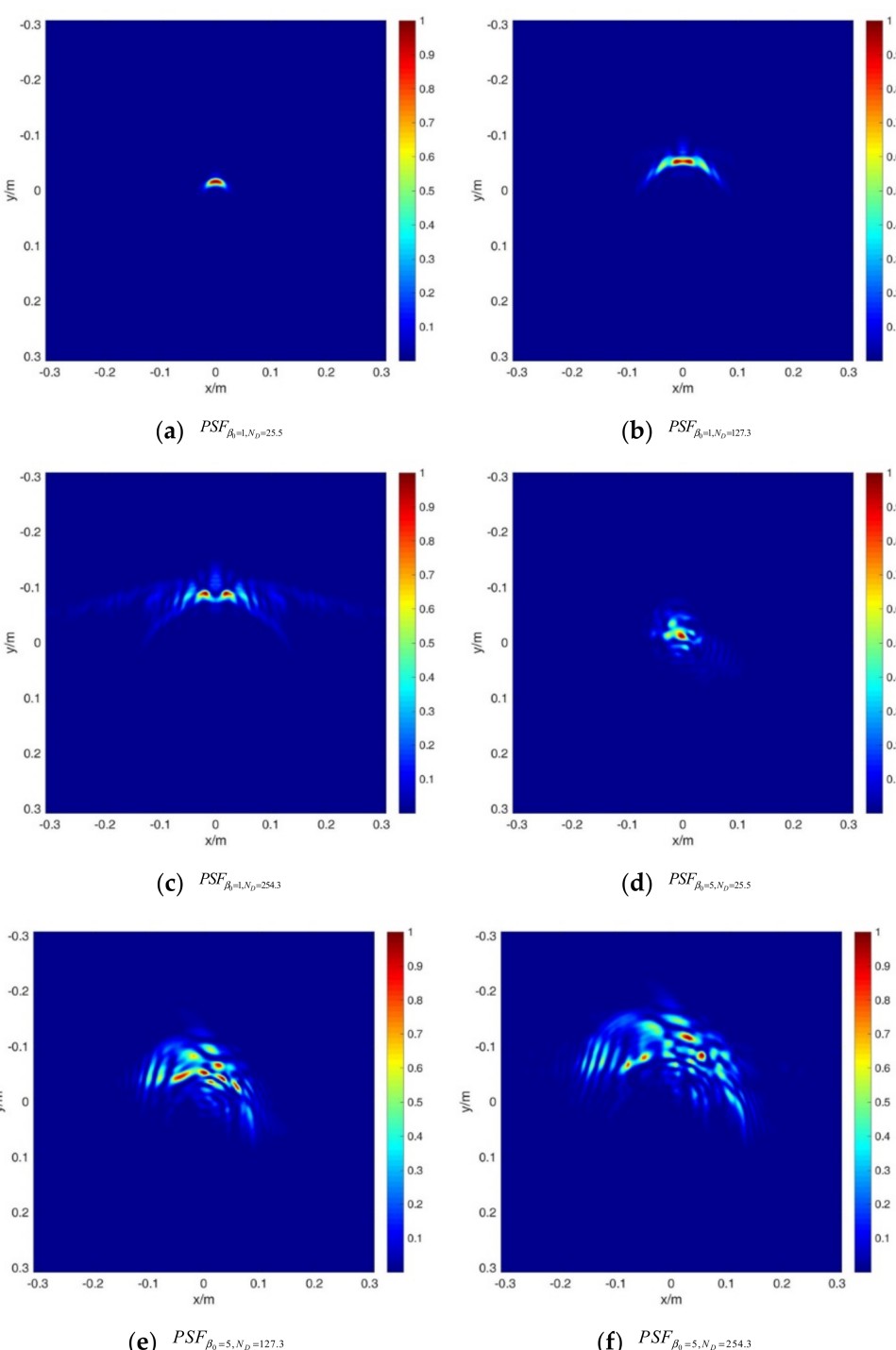

**Figure 5.** PSF of the system considering aberration and thermal blooming effect for (**a**) $PSF_{\beta_0=1,N_D=25.5}$, (**b**) $PSF_{\beta_0=1,N_D=127.3}$, (**c**) $PSF_{\beta_0=1,N_D=254.3}$, (**d**) $PSF_{\beta_0=5,N_D=25.5}$, (**e**) $PSF_{\beta_0=5,N_D=127.3}$, and (**f**) $PSF_{\beta_0=5,N_D=254.3}$.

*3.3. Verification of the Fast Calculation Model with PSF Convolution*

3.3.1. Results for Turbulence-Only Effects

First of all, we consider the turbulence effect to verify the calculation results of the far-field intensity of light propagation by the fast calculation with *PSF* convolution. The comparison results between the fast calculation results of $PSF_{FC}$ and the results of 4D wave optics simulation program are presented. The far-field intensity distribution results of the four propagation scenarios (S1, S2, S3 and S4) under different turbulence, jitter, and system

aberration conditions are shown in Figure 6. The left column is the calculation result of the 4D wave optics simulation program and the right column is the theoretical calculation result of $PSF_{FC}$. In the $PSF_{FC}$ calculation, Equation (3) is used to convolute the $PSF$ with different effects, as seen in Table 3. As can be seen from Figure 6, the general shape of light facula generated by $PSF_{FC}$ and those by the 4D wave optics simulation program compare relatively well in all scenarios. There is a small deviation in the position of the center of mass for the turbulence effect.

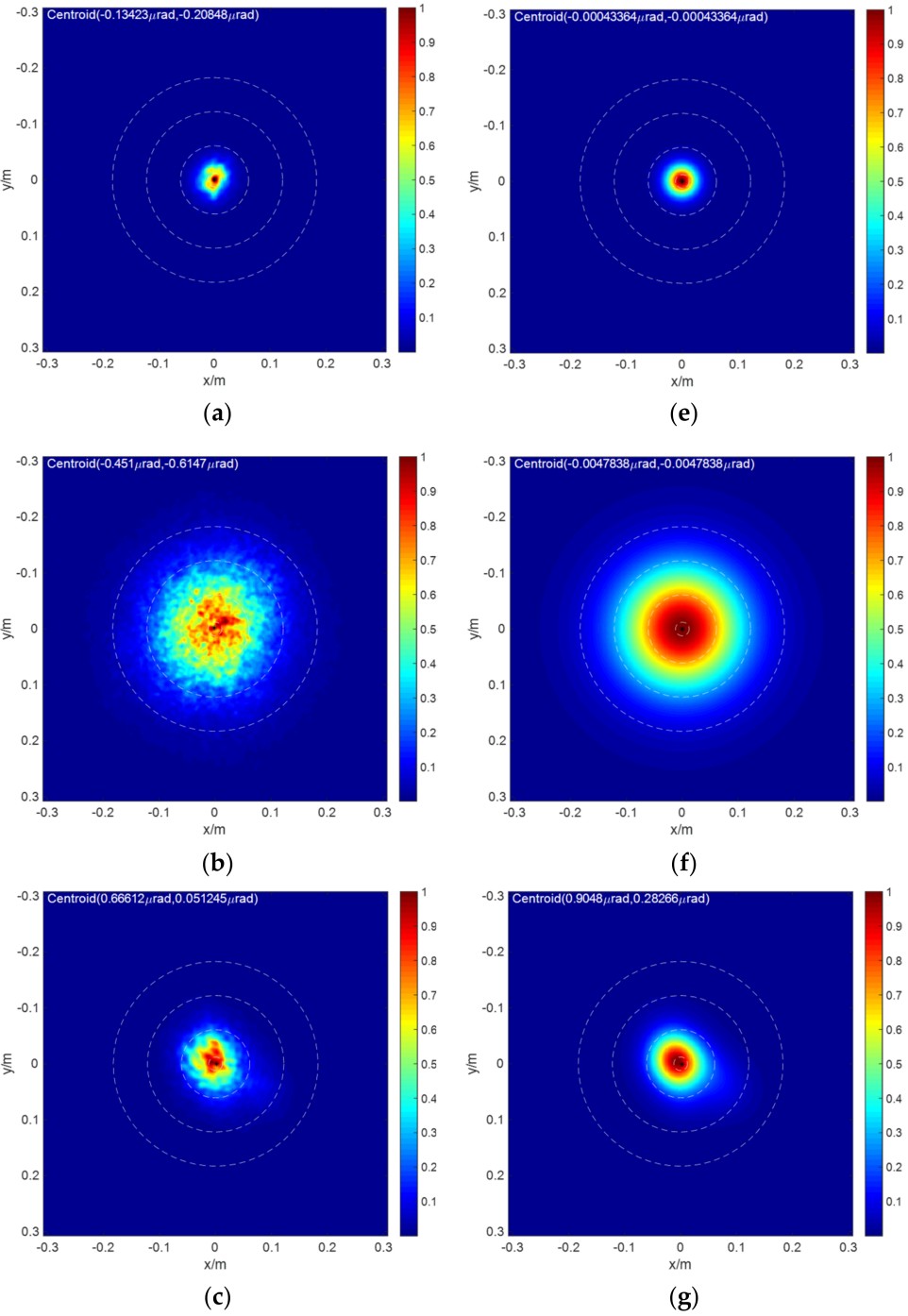

**Figure 6.** *Cont.*

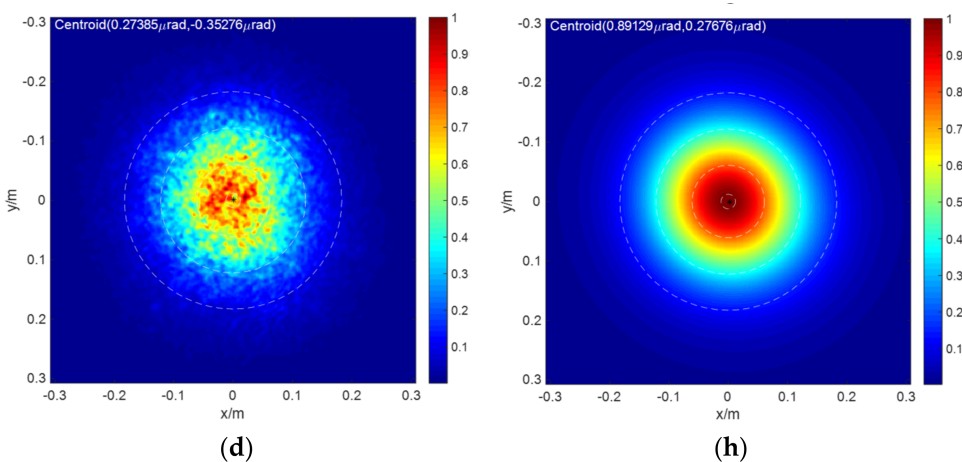

**(d)**　　　　　　　　　　　　　　　　　　　　　　　**(h)**

**Figure 6.** 4D wave optics simulation results (**a–d**) and the far-field intensity distribution for fast calculation results (**e–h**) with S1, S2, S3, and S4 scenario pairs presented top to bottom, respectively. The center to outermost calibration rings around the central propagation axis in this and subsequent figures have arbitrarily sized diameters to represent the diffraction-limit, 5.0 times diffraction limit, 10.0 times diffraction limit, and 15.0 times diffraction limit, respectively.

**Table 3.** Fast calculation with PSF convolution process.

| Scenario | Parameters | Fast Calculation |
|:---:|:---:|:---:|
| S1 | $\beta_0 = 1, D/r_0 = 4.0$ | $PSF_{FC} = PSF_{\beta_0=1} \otimes PSF_{D/r_0=4.0}$ |
| S2 | $\beta_0 = 1, D/r_0 = 16.1$ | $PSF_{FC} = PSF_{\beta_0=1} \otimes PSF_{D/r_0=16.1}$ |
| S3 | $\beta_0 = 5, jitter = 5\mu rad, D/r_0 = 4.0$ | $PSF_{FC} =$ $PSF_{\beta_0=5} \otimes PSF_{jitter=5\mu rad} \otimes PSF_{D/r_0=4.0}$ |
| S4 | $\beta_0 = 5, jitter = 10\mu rad, D/r_0 = 16.1$ | $PSF_{FC} =$ $PSF_{\beta_0=5} \otimes PSF_{jitter=10\mu rad} \otimes PSF_{D/r_0=16.1}$ |

In addition to the general qualitative comparison of the relative spot size and shape, we also quantitatively evaluated the far-field irradiance distribution with the normalized encircle-axis energy ($Enc_{axis}$) and normalized encircle-peak energy ($Enc_{peak}$) as the evaluation metric. The change curve comparison between the fast calculation results and the 4D wave optics simulation results of $Enc_{axis}$ and $Enc_{peak}$ with facula expansion multiple $\beta$ under S1, S2, S3, and S4 scenarios are shown in Figure 7. It can be seen from the figures that, under the turbulence effect, the convolution calculation method proposed in this paper is in good agreement with the results of 4D optic wave simulation for HEL laser propagation. Generally speaking, the error of $Enc_{axis}$ is a little larger than that of $Enc_{peak}$.

Figure 8 shows the relative error between the fast calculation results and 4D wave optics simulation results for $Enc_{axis}$ and $Enc_{peak}$ under S1, S2, S3 and S4 scenarios. For all scenarios, the fast calculation results are smaller, which is a slightly more conservative estimate of performance than wave optics simulation. According to the calculation parameters of the scenarios, the size of the Airy spot is 1.22 cm. For $\beta = 5, 10, 15$ (spot radius represent 6.1 cm, 12.2 cm, and 18.3 cm), the errors are within $-15\%$, $-10\%$, and $-5\%$, respectively.

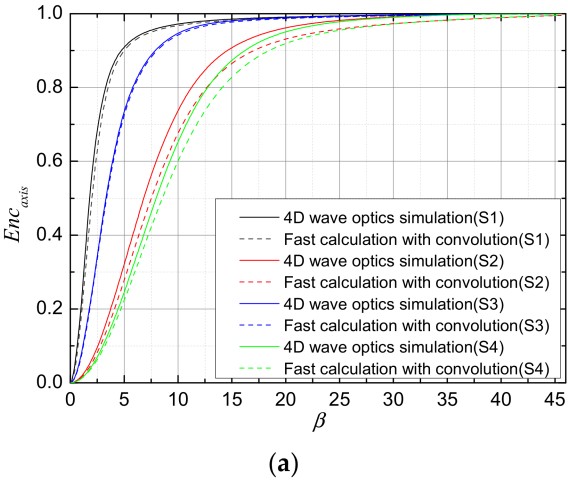
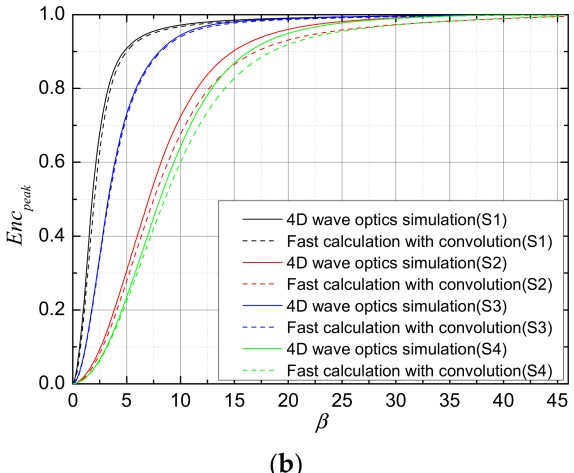

**Figure 7.** The change curve comparison between the fast calculation results and 4D wave optics simulation results of (**a**) $Enc_{axis}$ and (**b**) $Enc_{peak}$ with facula expansion multiple $\beta$ under S1, S2, S3, and S4 scenarios.

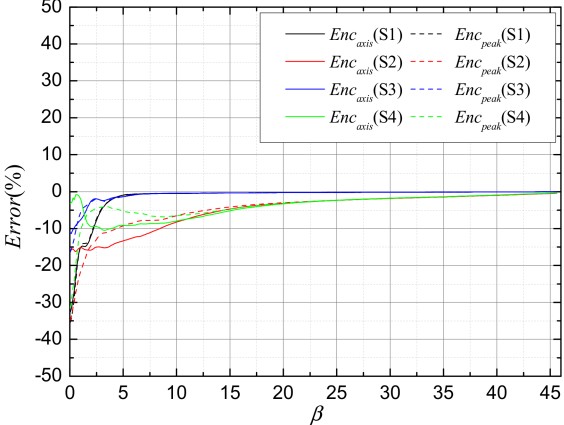

**Figure 8.** The relative error between the fast calculation results and 4D wave optics simulation results for $Enc_{axis}$ and $Enc_{peak}$ under S1, S2, S3, and S4 scenarios.

### 3.3.2. Results for Turbulence and Thermal Blooming Effects

We then consider the combined effects of turbulence and thermal blooming. Figure 9 shows the comparison between the theoretical calculation results of $PSF_{FC}$ and the 4D wave optics simulation program calculation results under different initial aberration and thermal distortion parameters for three propagation scenarios (S5, S6, and S7). Similarly, $PSF_{FC}$ is obtained by convoluting the $PSF$ with different effects, as seen in Table 4. In Figure 9, the left column shows the calculation result of the 4D wave optics simulation program, and the right column shows the fast calculation result of $PSF_{FC}$. It can be seen from the figures that, when the thermal blooming effect is weak, the calculation results of the two methods are in good agreement. With the increase of thermal distortion parameters, the light intensity distribution calculated by the two methods is different. The light facula energy calculated by the OTF method is more dispersed. This difference is mainly due to the assumption that the turbulent phase and the thermal blooming phase are independent in the transfer function calculation method. In actual conditions, the turbulent thermal blooming effect is interactive. However, from the point of view of rapid evaluation, the results of the two methods are in good agreement with each other, both from the far-field irradiance pattern and the energy distribution around the ring. With the increase of thermal blooming effect, the difference in the spot shape becomes larger.

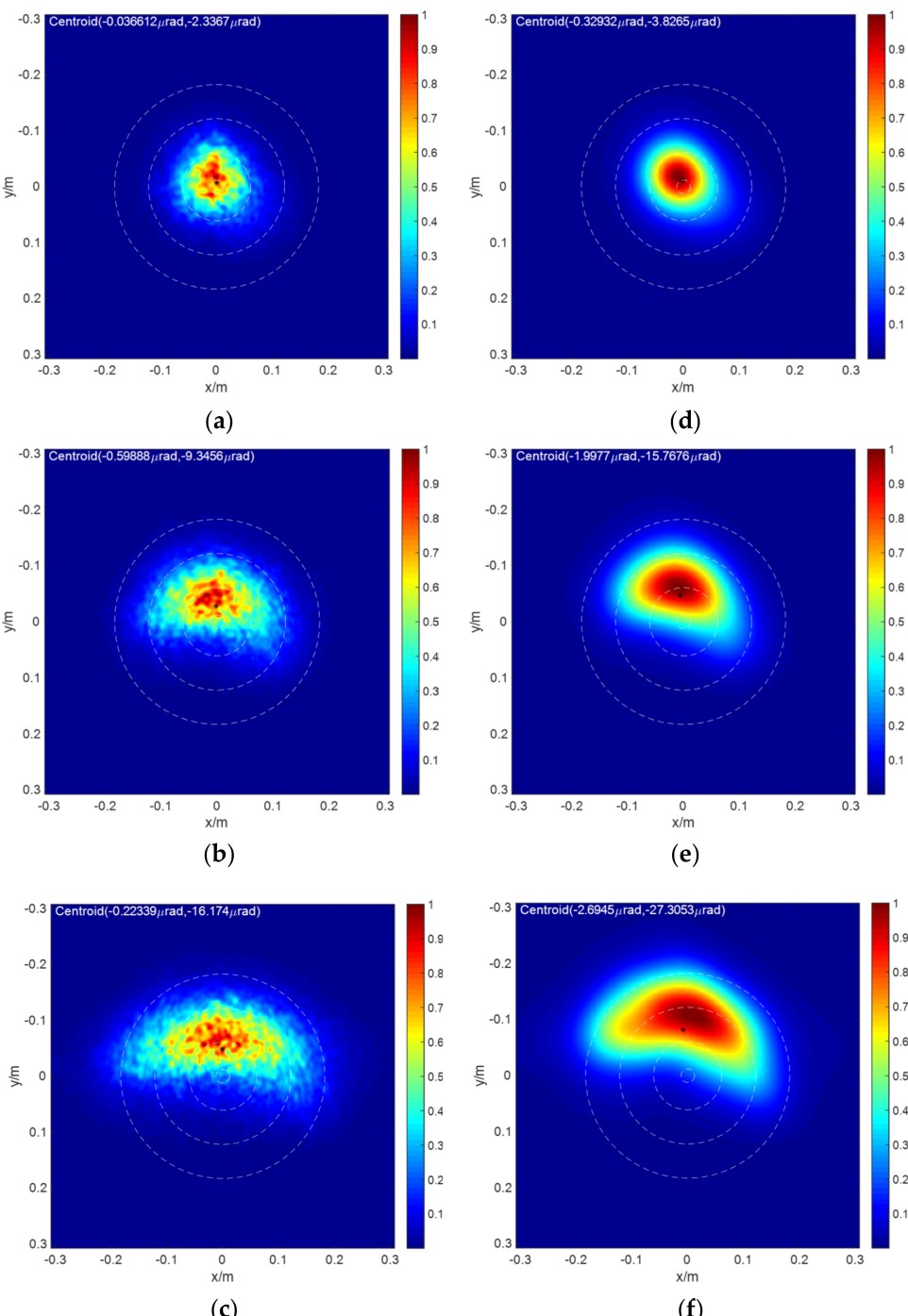

**Figure 9.** 4D wave optics simulation results (**a–c**) and the far-field intensity distribution for fast calculation results (**d–f**) with S5, S6, and S7 scenario pairs presented top to bottom, respectively.

**Table 4.** Fast calculation with PSF convolution process.

| Scenario | Parameters | Fast Calculation |
|---|---|---|
| S5 | $\beta_0 = 5, jitter = 5\mu rad, D/r_0 = 6.13, N_D = 25.5$ | $PSF_{FC} = PSF_{\beta_0=5} \otimes PSF_{jitter=5} \otimes PSF_{D/r_0=6.13} \otimes PSF_{N_D=25.5}$ |
| S6 | $\beta_0 = 5, jitter = 5\mu rad, D/r_0 = 6.13, N_D = 127.3$ | $PSF_{FC} = PSF_{\beta_0=5} \otimes PSF_{jitter=5} \otimes PSF_{D/r_0=6.13} \otimes PSF_{N_D=127.3}$ |
| S7 | $\beta_0 = 5, jitter = 5\mu rad, D/r_0 = 6.13, N_D = 254.3$ | $PSF_{FC} = PSF_{\beta_0=5} \otimes PSF_{jitter=5} \otimes PSF_{D/r_0=6.13} \otimes PSF_{N_D=254.3}$ |

Figure 10 shows the change curve comparison between the fast calculation results and 4D wave optics simulation results of $Enc_{axis}$ and $Enc_{peak}$ with facula expansion multiple $\beta$ under S5, S7, and S7 scenarios. Figure 11 gives the relative error between the fast calculation results and the 4D wave optics simulation results for $Enc_{axis}$ and $Enc_{peak}$. With the increase of thermal distortion parameter, the difference of $Enc_{axis}$ increases, but the difference in $Enc_{peak}$ is not significant. For $\beta = 5, 10, 15$, the errors of $Enc_{axis}$ are within $-50\sim10\%$, and the errors of $Enc_{peak}$ are within $-25\sim5\%$.

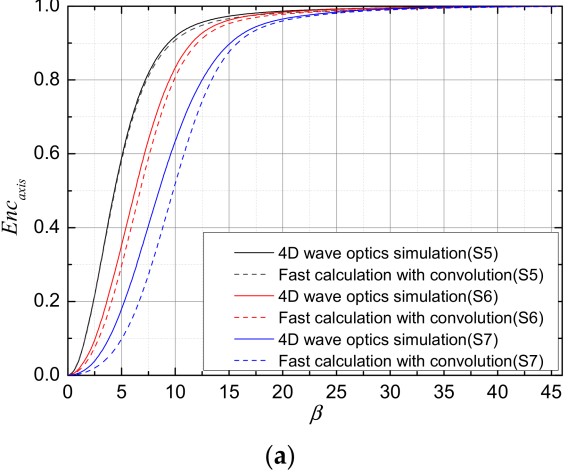 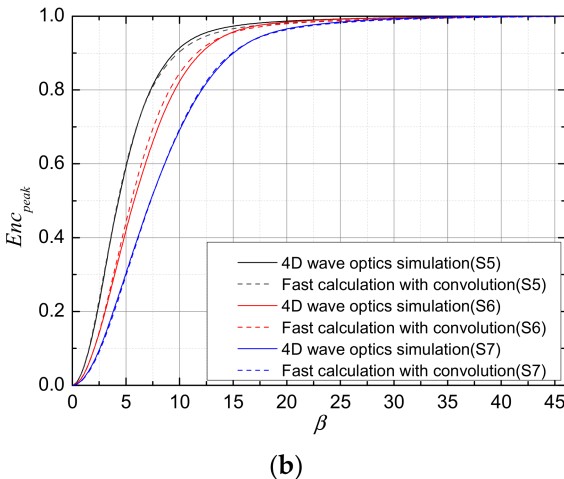

(**a**)                                                                                                    (**b**)

**Figure 10.** The change curve comparison between the fast calculation results and 4D wave optics simulation results of (**a**) $Enc_{axis}$ and (**b**) $Enc_{peak}$ with facula expansion multiple $\beta$ under S5, S6, and S7 scenarios.

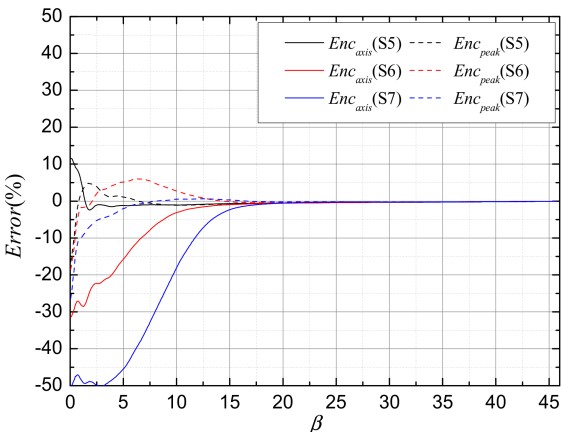

**Figure 11.** The relative error between the fast calculation results and 4D wave optics simulation results for $Enc_{axis}$ and $Enc_{peak}$ under S5, S6, and S7 scenarios.

### 3.3.3. Results Verification of Uplink Transmission Scenario Results

In order to further verify the performance of the convolution fast calculation method, we further study the scene of transmission at a certain elevation angle. The laser is transmitted 10 km obliquely upward at an elevation angle of 30 degrees from the ground. For a horizontal transmission path, the turbulent refractive index structure parameter is usually assumed to be a constant. For a slant transmission path, a turbulence profile model is required. There are already many atmospheric turbulence profile models [35–38]. We chose the Hufnagel-Valley model to begin our simulation because it is well known and often used and has two parameters to adapt to atmospheric conditions; one is the surface refractive index structure parameter $C_n^2(0)$ (m$^{-2/3}$), and the other is root mean square wind speed $U$ (m/s). This model calculates $C_n^2$ using the following equation [35,36]:

$$C_n^2(h) = 8.148 \times 10^{-26} U^2 h^{10} \exp(-h) + 2.700 \times 10^{-16} \exp(-h/1.5) + C_n^2(0) \exp(-10h) \tag{15}$$

where $h$ is height in kilometers. For the H-V5/7 model, $C_n^2(0) = 1.7 \times 10^{-14}$ m$^{-2/3}$ and $U = 21$ m/s. The wind-speed profile we adopt is the Bufton wind model [39,40], that is:

$$V(h) = V_g + 30 \exp(-((h-9.4)/4.8)^2) \tag{16}$$

where 30 is wind speed at the 200 hPa level, 9.4 is the height of the upper boundary of the tropospehre, 4.8 is the thickness of the tropopause, and the ground wind speed parameter $V_g$ is usually taken to be 5 m/s. The absorption coefficient and extinction coefficient is shown in Figure 12. $PSF_{FC}$ is obtained by convoluting the $PSF$ with different effects, as seen in Table 5.

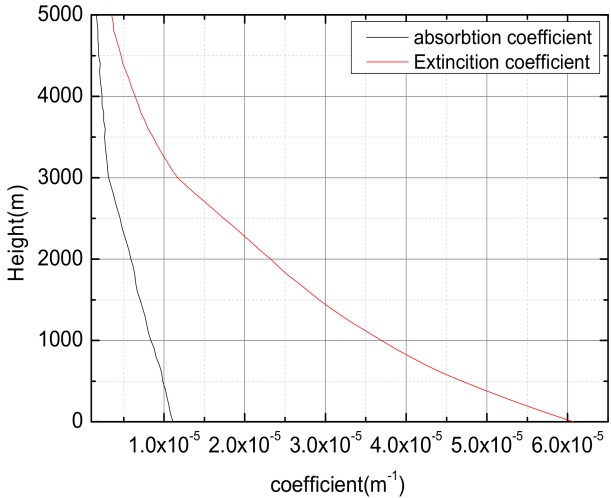

**Figure 12.** The absorption coefficient and extinction coefficient vary with height.

**Table 5.** Fast calculation with PSF convolution process.

| Scenario | Parameters | Fast Calculation |
|---|---|---|
| S8 | $\beta_0 = 5$, $jitter = 5\mu rad$, D/$r_0$ = 3.37, $N_D$ = 37.7 | $PSF_{FC} = PSF_{\beta_0=5} \otimes PSF_{jitter=5} \otimes PSF_{D/r_0=3.37} \otimes PSF_{N_D=37.7}$ |
| S9 | $\beta_0 = 5$, $jitter = 5\mu rad$, D/$r_0$ = 3.37, $N_D$ = 149.4 | $PSF_{FC} = PSF_{\beta_0=5} \otimes PSF_{jitter=5} \otimes PSF_{D/r_0=3.37} \otimes PSF_{N_D=149.4}$ |
| S10 | $\beta_0 = 5$, $jitter = 5\mu rad$, D/$r_0$ = 3.37, $N_D$ = 261.4 | $PSF_{FC} = PSF_{\beta_0=5} \otimes PSF_{jitter=5} \otimes PSF_{D/r_0=3.37} \otimes PSF_{N_D=261.4}$ |

We consider the combined effects of turbulence and thermal blooming. Figure 13 shows the comparison between the theoretical calculation results of $PSF_{FC}$ and the 4D wave optics simulation program calculation results under different thermal distortion parameters for S8, S9, and S10 scenarios. In Figure 13, the left column is the calculation result of the 4D wave optics simulation program, and the right column is the fast calculation result of $PSF_{FC}$. It can be seen from the figures that, when the thermal blooming effect is weak, the calculation results of the two methods are in good agreement. With the increase of thermal distortion parameters, the light intensity distribution calculated by the two methods is different. The light facula energy calculated by the OTF method is more dispersed. Again, we get the same conclusion.

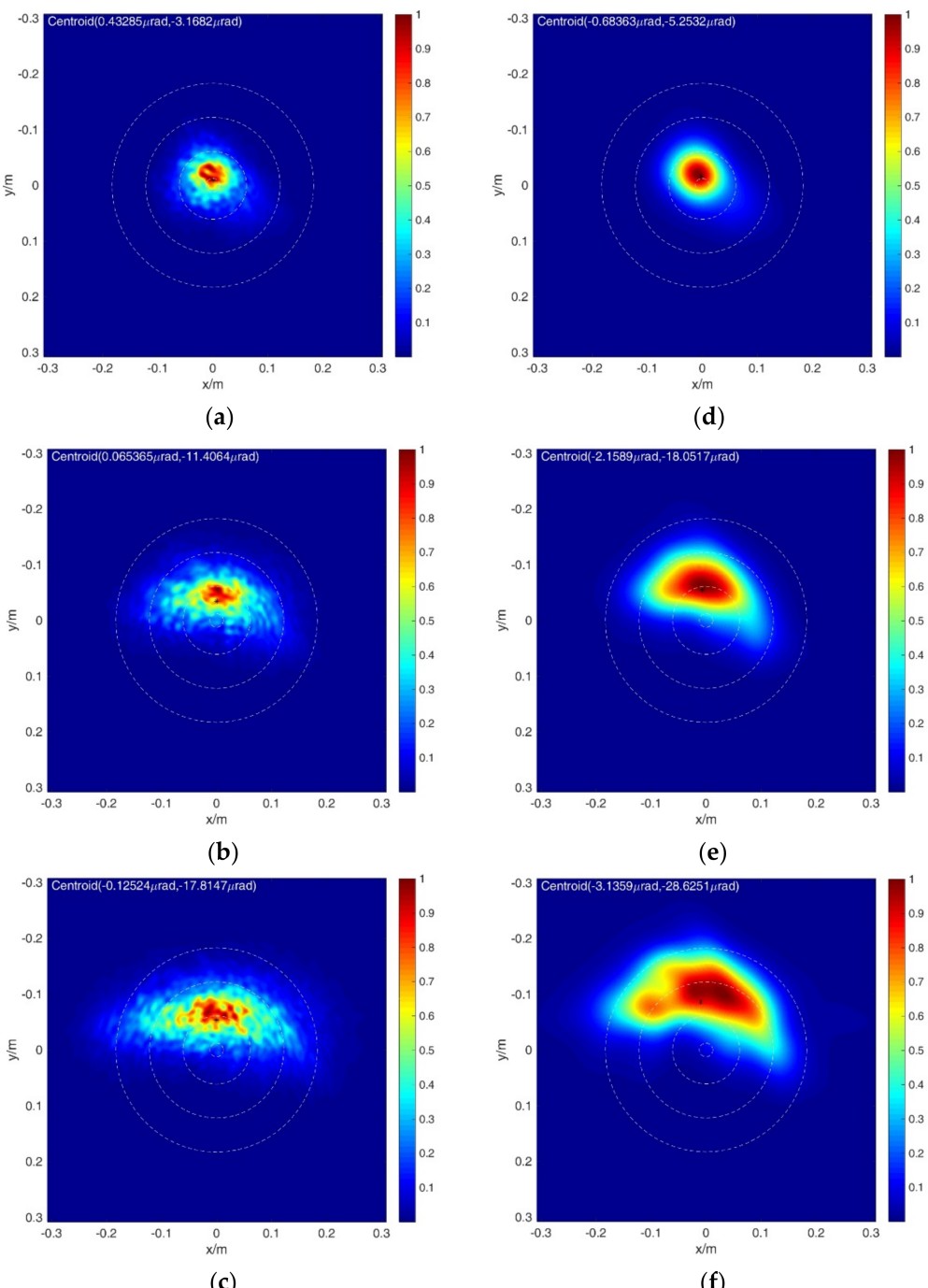

**Figure 13.** 4D wave optics simulation results (**a**–**c**) and the far-field intensity distribution for fast calculation results (**d**–**f**) with S8, S9, and S10 scenario pairs presented top to bottom, respectively.

Figure 14 shows the change curve comparison between the fast calculation results and the 4D wave optics simulation results of $Enc_{axis}$ and $Enc_{peak}$ with facula expansion multiple $\beta$ under S8, S9, and S10 scenarios. Similarly, when the thermal effect is weak, the fast calculation results are in good agreement with the 4D wave optics simulation results. With the increase of the thermal distortion parameter, the difference in $Enc_{axis}$ increases, but the difference in $Enc_{peak}$ is not significant. Figure 15 gives the relative error between the fast calculation results and the 4D wave optics simulation results for $Enc_{axis}$ and $Enc_{peak}$. For $\beta = 5, 10$, the errors of $Enc_{axis}$ are within $-60 \sim -20\%$, and the errors of $Enc_{peak}$ are within $-10 \sim 5\%$. For $\beta = 15$, the errors in $Enc_{axis}$ and $Enc_{peak}$ are all negligible.

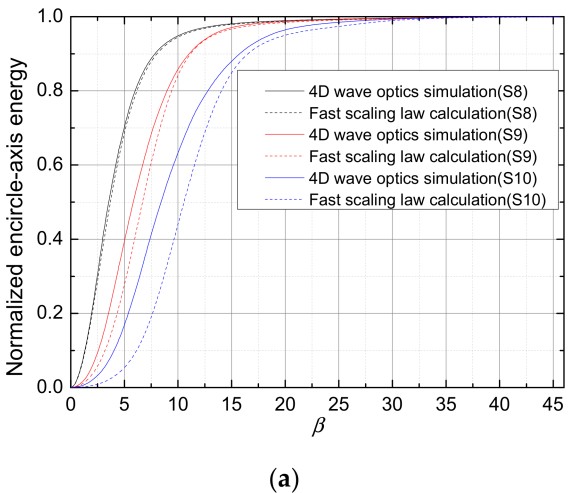 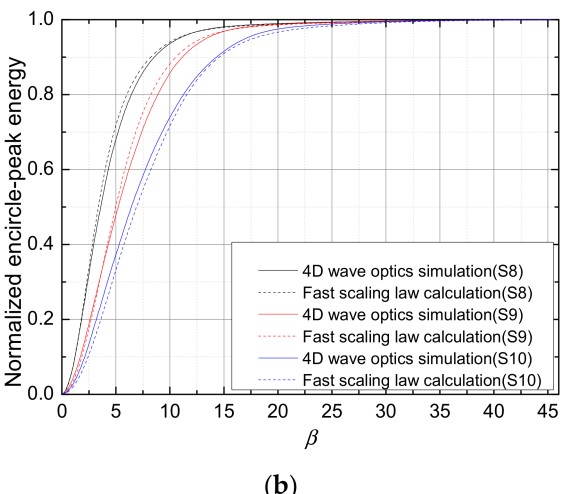

(**a**)  (**b**)

**Figure 14.** The change curve comparison between the fast calculation results and 4D wave optics simulation results of (**a**) $Enc_{axis}$ and (**b**) $Enc_{peak}$ with facula expansion multiple $\beta$ under S8, S9, and S10 scenarios.

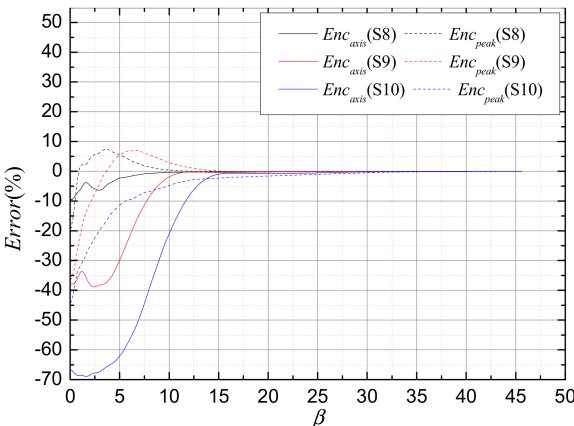

**Figure 15.** The relative error between the fast calculation results and 4D wave optics simulation results for $Enc_{axis}$ and $Enc_{peak}$ under S8, S9, and S10 scenarios.

## 4. Conclusions

A fast calculation method of far-field intensity distribution with *PSF* convolution for thermal blooming and turbulence effects on HEL propagation is proposed in this paper. We calculate the far-field intensity of the laser propagation system from the point of view of the point source imaging system to calculate the PSF. A series of effects, such as infinite aperture, platform jitter, atmospheric turbulence, and thermal blooming, are all regarded as optical transfer functions of imaging systems. When turbulence effect, thermal blooming effect, and jitter phase are considered to be independent of each other, and when the thermal blooming effect of continuous laser reaches a steady state, the aberration caused by the thermal blooming effect will not change; the fast calculation results of $PSF_{FSL}$ can be obtained by convoluting the *PSF* of each effect. In the study, the results calculated by this method under different aberration, turbulence, and thermal conditions for the horizontal transmission scene and uplink transmission scene are verified by those calculated by the 4D wave optics simulation program. The results show that the method is in good agreement with the calculation results of the 4D wave optics simulation program under the linear effect, and there are some differences between the two under the condition of the turbulent thermal blooming effect, but as a rapid evaluation method, it has a high accuracy. This difference mainly comes from the assumption that the interaction between turbulence and thermal blooming is independent. The fast calculation model and method mentioned in this paper can provide a reference for the engineering application of the rapid evaluation

for HEL propagation. In particular, it has great practical value for HEL evaluations, which require large amounts of far-field intensity information quickly. However, it is important to point out that the results calculated by the proposed method are relatively accurate when the thermal blooming effect is weak, but when the thermal blooming effect is strong and the interaction between turbulence and thermal blooming is obvious, the deviation of the results will increase significantly.

For further work, we will expand the assessment capability of the fast calculation method of far-field intensity distribution with *PSF* convolution, such as considering other shapes of aperture and the beam passing through other random media, such as rain, fog aerosols, and ocean turbulence.

**Author Contributions:** Conceptualization, P.Z. and H.M.; methodology, P.Z.; software, P.Z. and J.Z.; validation, C.F., C.Q., and H.M.; formal analysis, P.Z. and C.F.; investigation, C.Q. and H.L.; resources, P.Z.; data curation, H.M., W.Z., and X.L.; writing—original draft preparation, H.M.; writing—review and editing, H.M.; visualization, H.M. and H.L.; supervision, C.F.; project administration, J.Z.; funding acquisition, H.M. All authors have read and agreed to the published version of the manuscript.

**Funding:** This research was funded by National Natural Science Foundation of China under Grant 61905002 and Natural Science Research Projects in Anhui Universities under Grant KJ2019A0210.

**Institutional Review Board Statement:** Not applicable.

**Informed Consent Statement:** Not applicable.

**Data Availability Statement:** Data sharing is not applicable to this article.

**Conflicts of Interest:** The authors declare no conflict of interest.

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
