# Peer review of "A Fast Calculation Method of Far-Field Intensity Distribution with Point Spread Function Convolution for High Energy Laser Propagation"

_applsci, doi:10.3390/app11104450_

Round 1
Reviewer 1 Report
The manuscript is devoted to the development of a fast calculation method of far-field intensity distribution with point spread function convolution and uses the 4 D simulation programm. The manuscript has interesting topic, written in a scientific language, but it is not free from disadvantages.
a) The authors propose the fast calculation method of far-field intensity distribution with point spread function convolution However, detailed studies of the speed operation of the method have not been performed and the time frames for the operation of the method for different input parameters are not indicated.
b) The study is devoted to simulation (not experiments) and requires at least a small comparison with real data in the atmosphere.
c) The authors give formula 13 with cn2 (h) but do not indicate the height profiles cn2 (h). In other words, did the authors simulate turbulence on a horizontal path (where the average value of cn2 (h) can be used and then Formula 13 is simplified and should be written in a different form) or on a vertical path (including the model case of an inclined path) and then it is necessary to describe the dependencies used, namely vertical profiles cn2 (h) used in calculating r0 (https://arxiv.org/pdf/1101.3924.pdf, 10.1134/S1024856019020076, https://arxiv.org/abs/1001.1304) .
d) It is necessary to substantiate the selected values of cn2 in Table 2. These values ​correspond to the regimes of weak turbulence (D / r0 = 4) or strong turbulence (D / r0 = 4 16.1). This (regimes) must be indicated. The cn2 (average?) values used in Table 2 correspond to optical turbulence strength in the surface layer of the atmosphere (https://doi.org/10.1364/OE.24.020424, https://doi.org/10.3390/atmos10110661)
e) In the conclusions, it is necessary to note clearly the advantages and disadvantages of the proposed method.
f) It is necessary to improve the introduction and note the relevance of the development of similar methods, for example, for astronomy. The PSF calculation is essential for modeling the optical distortions of high-speed adaptive optics systems. It is possible (at the point of view of the authors) to provide additional references (https://iopscience.iop.org/article/10.1086/343217, https://arxiv.org/abs/1908.02200, https://iopscience.iop.org/article/10.1086/505547, https://doi.org/10.1117/12.2205600).
To increase the significance of the research as a whole, the article should be considered in the context of a real turbulent atmosphere with rapidly changing Сn2(h).
Reviewer 2 Report
The manuscript reports on a fast calculation method of the far-field intensity distribution of high energy laser. The propagation process of high-energy laser is modeled as the imaging process of point source. The high energy laser propagation is simulated for several atmospheric effects that can seriously distort the beam, namely turbulence effect, thermal blooming effect, laser beam aberration, and platform jitter. Finally, they made a thorough comparison between their fast calculation method and the 4D wave optics simulation method.
In general, PSF convolutions are used in many fields of physics including optical imaging, optical communication, etc. Most of their applications require real-time information processing, and fast PSF calculation methods can greatly enhance their performance.
The manuscript is mostly well written, and the results presented are suitable for the publication in the Journal of Applied Sciences. I have a minor comment regarding the frequently used long sentences in the text which can sometimes confuse the reader. Besides, I have one more general comment regarding the PSF profiles and their comparison (figures 6-11). In general, we can see that the fast calculation method results a speckled PSF profiles which is absent in 4D wave optics simulations. I could not find a clear explanation in the text why this effect is observed and what can be the major consequences of this effect in the real-world applications.
Round 2
Reviewer 1 Report
The article is very interesting and may be recommended for publication after the shortcomings eliminating. Nevertheless, I consider it necessary to note the following aspects. The authors pointed out that «the simulated transmission path is horizontal, and the turbulence refractive structure parameter is set as a constant (line 206). The phrase is incomplete as the authors consider different turbulence regimes, which are characterized by different r0. Maybe the authors mean constancy of Cn2 for each scenario: the turbulence refractive structure parameter is a statistical characteristic of the turbulence strength with at least minute averaging, but highly variable over time. On the other hand, this value is often called a constant, referring to the turbulence spectrum. In the case of a horizontal path in which the refractive index structure parameter Cn 2 is essentially constant… This is not true at 3 km distance. We can just assume that the refractive index structure parameter Cn2 is constant at a horizontal path. This is an assumption, not a statement. The spatial distribution of Cn2 changes significantly in the surface layer of atmosphere. I understand what the authors want to say, but it is still necessary to formulate more precisely. Authors pointed out that the Cn2 of the near-surface boundary layer is usually between 10-15 and 10-13 m^-2/3. The optical turbulence strength of our simulated is within this range. This should be explained in the text of the article. The calculation methods of Cn2 are different. Obviously, this is the input model parameter in your study. It is necessary to provide references to the Cn2 range of changes in the low part of boundary layer: -Statistics of the Optical Turbulence from the Micrometeorological Measurements at the Baykal Astrophysical Observatory Site /Atmosphere (https://doi.org/10.3390/atmos10110661)- Otakar Jicha, Pavel Pechac, Stanislav Zvanovec, Martin Grabner, Vaclav Kvicera Long-term measurements of refractive index structure constant in atmospheric boundary layer / Proceedings Volume 8535, Optics in Atmospheric Propagation and Adaptive Systems XV; 853509 (2012) https://doi.org/10.1117/12.974422
Section 3.3.3
The authors provide a formula 16 with numerical coefficients but they don't explain these coefficients. it is necessary to explain these coefficients. For example, 30 is wind speed at the 200 hPa level, 9.4 is the height of the upper boundary of the tropospehre, 4.8 is the thickness of the tropopause. Also,if you're talking about HV-profiles, it is advisable to refer to other models and results as the HV model describes better the structure of turbulence above the atmospheric boundary layer (higher than 1.5-3 km):
-https://www.mdpi.com/2073-4433/10/9/499
- Trinquet H., Vernin J. A statistical model to forecast the profile of the index structure constant $${C_N^2} / Environmental Fluid Mechanics 7(5):397-407
The authors used a simple HV model within Cn2 is changed with distance (height). This model may be adapted to atmospheric conditions. I believe that it is necessary to explain this in the text.
I would like to emphasize once again that the article is quite interesting and important!, and can be published after the shortcomings are removed. At the same time, I would like to see the answers to the comments to be discussed (so the status is major) I hope this is the last iteration.
Author Response
Thank you for your careful and professional review. We have revised all the above aspects and marked them in red.
Round 3
Reviewer 1 Report
I have only one comment. Line 207. I suggest to remove "near the surface" as there are two words "surface" in this sentence.